# UNDERSTANDING ZERO-SHOT ADVERSARIAL ROBUSTNESS FOR LARGE-SCALE MODELS

**Chengzhi Mao**[1]* **Scott Geng**[1]* **Junfeng Yang**[1] **Xin Wang**[2] **Carl Vondrick**[1]
Columbia University[1], Microsoft Research[2]

## ABSTRACT

Pretrained large-scale vision-language models like CLIP have exhibited strong generalization over unseen tasks. Yet imperceptible adversarial perturbations can significantly reduce CLIP's performance on new tasks. In this work, we identify and explore the problem of *adapting large-scale models for zero-shot adversarial robustness*. We first identify two key factors during model adaption—training losses and adaptation methods—that affect the model's zero-shot adversarial robustness. We then propose a text-guided contrastive adversarial training loss, which aligns the text embeddings and the adversarial visual features with contrastive learning on a small set of training data. We apply this training loss to two adaption methods, model finetuning and visual prompt tuning. We find that visual prompt tuning is more effective in the absence of texts, while finetuning wins in the existence of text guidance. Overall, our approach significantly improves the zero-shot adversarial robustness over CLIP, seeing an average improvement of 31 points over ImageNet and 15 zero-shot datasets. Our code and model is available at **github.com/cvlab-columbia/ZSRobust4FoundationModel**.

## 1 INTRODUCTION

Large-scale models trained on vision and language data—also known as foundation models— have emerged as a universal backbone for tackling many recognition problems in computer vision (Jia et al., 2021; Radford et al., 2021), graphics (Ramesh et al., 2022) and robotics (Ahn et al., 2022). One of the key advantages of foundation models is zero-shot generalization, where the models use just a single textual description to recognize new visual categories with high accuracy. Since those large-scale models are powerful, they will continue to be used in many critical applications, where it is important to make them reliable. However, robustness under adversarial examples remains a challenge, where an imperceptible pattern can be combined with the image to cause recognition failures (Croce & Hein, 2020; Carlini & Wagner, 2017; Dong et al., 2018; Szegedy et al., 2013; Moosavi-Dezfooli et al., 2016), where attack on foundation models can consequently corrupt the downstream applications.

Due to the importance of this problem, there is a large literature that investigates adversarial robustness for neural networks. The most common approach for adversarial defense is to learn the model through adversarial training (Madry et al., 2018; Mao et al., 2019; Szegedy et al., 2013; Pang et al., 2020; Rice et al., 2020; Uesato et al., 2019), which involves augmenting the training set with mined adversarial examples that fool the image classifier. Adversarial training has been validated to improve robustness on the task that the mined examples come from, but it often comes at a cost of generalization (Stutz et al., 2019; Su et al., 2018; Pedraza et al., 2021). However, our world is vast and naturally open, and only evaluating adversarial robustness on the learned tasks is limited. *Can we achieve zero-shot transferability for adversarial robustness, even if the model has never been trained on the unknown tasks?*

In this paper, we study this important yet under-explored problem, *zero-shot adversarial robustness* of large-scale vision-language models. We start our investigation with the state-of-the-art CLIP model (Radford et al., 2021), which has been shown to be effective in zero-shot recognition tasks. We find that simply adding an imperceptible vector to the image ($\leq 1/255$) can subvert

---

*Equal contribution

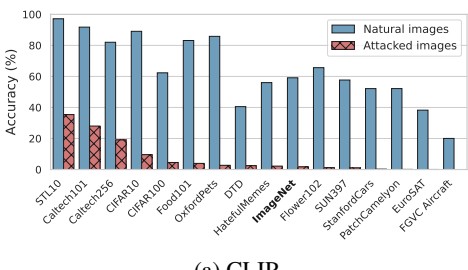 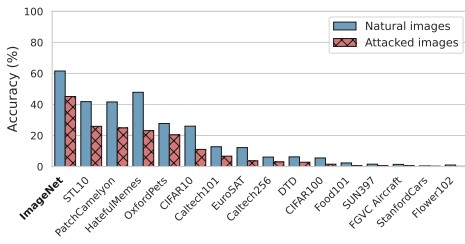

|(a) CLIP|(b) Adversarially Finetuned CLIP|

Figure 1: (a, left) Despite CLIP's high performance on zero-shot image recognition tasks, it remains vulnerable when the input images are constructed adversarially. (b, right) Standard adversarial training improves robustness on the trained task (ImageNet), but comes at the expense of its zero-shot capability. Our paper studies how to adapt CLIP to achieve adversarial robustness on zero-shot tasks.

CLIP's prediction (see Figure 1a). If we follow the standard adversarial training defense paradigm (Madry et al., 2018; Rice et al., 2020) to finetune CLIP on the ImageNet (Deng et al., 2009b) training set, we observe that the adapted CLIP has improved adversarial robustness on the ImageNet validation set, but comes at the cost of significantly reduced accuracy on unseen datasets and classes (Figure 1b). Standard adversarial training backfires on CLIP as it fails to retain the model's zero-shot generalization ability.

Adaptation methods and training objectives are the two major factors for adapting a large-scale model. First, besides finetuning the whole model, we seek an alternative adaptation method—visual prompt tuning—which adapts the *inputs* instead of the *parameters* of the model. Visual prompt tuning (VPT) is an emerging light-weight adaptation method (Bar et al., 2022; Bahng et al., 2022) that learns a visual prompt which is added to the input image, where we use visual prompt to instruct the model to be robust against adversaries. Second, we find that the standard adversarial training objective ignores the visual-language alignment in CLIP's pretrained representation space, causing the model to lose zero-shot capability. We then propose a text-guided contrastive adversarial training (TeCoA) loss, dubbed as Tekoa (tee·kow), which maximizes the similarity of the adversarial visual features and the correct text embeddings with contrastive learning. Since the adapted visual features continue to align well with the text features, the model adapted with TeCoA can maximally retain the original zero-shot generalization of CLIP while enjoying improved adversarial robustness.

We conduct an extensive evaluation on 15 zero-shot image datasets, offering a holistic study of the zero-shot adversarial robustness problem. This is especially important given that large-scale vision models are emerging as infrastructure and are deploying to critical applications. We find that the lightweight VPT is noticeably more effective than model finetuning when textual information is unavailable. When texts are used during adaptation, both VPT and finetuning using our TeCoA loss have drastically improved zero-shot adversarial robustness compared to baselines. Finetuning has higher gains than VPT as more parameters are tuned. Our best performing model with the TeCoA loss can improve adversarial robustness over CLIP by an average of 31% across the datasets. Our method also works on unlabeled images, allowing for better robustness with a large amount of unlabeled data. Our work establish a new and important benchmarket, zero-shot adversarial robustness, for future work to evaluate on. We release all models and code.

## 2 RELATED WORK

**Zero-Shot Generalization** aims to classify novel classes and tasks that are unseen during training (Palatucci et al., 2009; Lampert et al., 2009; Radford et al., 2021). Existing zero-shot methods often project visual features into semantic feature space (Frome et al., 2013; Akata et al., 2015; Romera-Paredes & Torr, 2015; Xie et al., 2019; Yu et al., 2018; Liu et al., 2019), or use generative methods to generate fake visual features of unseen classes from their semantic descriptions to train classifiers (Xian et al., 2018; Ni et al., 2019; Huang et al., 2019; Schonfeld et al., 2019; Verma et al., 2019; Liu et al., 2020). Recently, large-scale pretrained vision-language models (Radford et al., 2021; Jia et al., 2021) have shown outstanding zero-shot generalization ability on unseen tasks via text prompt engineering. Their adversarial robustness and its transferability, however, has not been studied in the zero-shot setting.

**Adversarial Robustness.** Adversarial attacks for image recognition find an additive vector on the input to maximize the cross-entropy loss, which is calculated from the model prediction and the ground truth one-hot label (Szegedy et al. (2013); Athalye et al. (2018); Carlini & Wagner (2017);

Kurakin et al. (2017); Papernot et al. (2015); Moosavi-Dezfooli et al. (2016)). Adversarial training (Madry et al., 2018) and its variants (Zhang et al., 2019; Rice et al., 2020), which train the model on generated adversarial examples, are effective in improving adversarial robustness on the task that they have been trained on. However, it is unclear whether this approach can improve robustness in the zero-shot scenario. To the best of our knowledge, Yucel et al. (2020) is so far the only work that studies the adversarial robustness of zero-shot learning models. Their setting is limited because it relies on predicting robust attributes, which may not be easily available for many tasks.

**Transferability of Robustness.** Mao et al. (2020) shows that adversarial robustness is transferable across tasks when multiple tasks are trained together. Salman et al. (2020) shows that adversarially robust models transfer better than their standard-trained counterparts when they are finetuned on other tasks. Chan et al. (2020) finds that matching a robust model's gradient can transfer robustness to a new model. Vaishnavi et al. (2022) proposes a low-cost method to transfer robustness to a new model on the same task with different architecture. However, the transferability of adversarial robustness to zero-shot tasks has not been investigated.

**Contrastive Learning** (Oord et al., 2018) has been used to train large-scale image-language models (Jia et al., 2021; Radford et al., 2021). Kim et al. (2020); Jiang et al. (2020) propose instance-wise adversarial perturbation and use a contrastive loss to align the features of clean examples and generated adversarial examples. Our method is the first one to introduce a cross-modal image-text contrastive loss in adversarial contrastive learning.

**Adapting Pretrained Models.** Linear probing and finetuning are the two major ways to adapt deep pretrained models. Recently, a more lightweight adaptation method, prompt tuning, has been proposed (Zhou et al., 2022c;b;a). Shu et al. (2022) shows that test-time optimization for text prompting helps generalization. Bar et al. (2022) combines target task and image inpainting to achieve zero-shot task inference. Jia et al. (2022); Sandler et al. (2022) used visual prompting to replace the finetuning procedure for large-scale models. Bahng et al. (2022) optimizes a visual prompt to increase the performance on the same task that the model finetunes the visual prompt on. Mao et al. (2021); Lawhon et al. (2022); Mao et al. (2022) find input prompt with self-supervised objective for robustness. Prompt tuning for continuous learning has also been proposed (Conder et al., 2022). Liu et al. (2022) proposes an amortized approach to use fewer iterations to adapt models. Wang et al. (2019) showed that it is useful in improving the performance for healthcare, and Liu et al. showed it can be used to adapt generative models for solving under constrained problems. However, adapting large-scale models for transferable zero-shot robustness, using methods such as finetuning or visual prompting, has not been investigated.

## 3 MODEL ADAPTATION FOR ZERO-SHOT ADVERSARIAL ROBUSTNESS

We first give background in Section 3.1 on adversarial attacks, adversarial training, and the problem setup. In Section 3.2, we discuss adaptation methods for adapting large-scale models for zero-shot adversarial robustness. Finally, Section 3.3, we discuss the effect of different training losses to motivate and then introduce our proposed text-guided contrastive adversarial training (TeCoA) loss.

### 3.1 BACKGROUND AND PROBLEM SETUP

Let $F_\theta(\cdot)$ be a deep model for image recognition parameterized by $\theta$. Given an input image $\mathbf{x}$, the model produces a representation $\hat{\mathbf{y}} = F_\theta(\mathbf{x})$. For image classification, a standard model learns to minimize the cross-entropy loss, $\mathcal{L}(\mathbf{x}, \mathbf{y}) = H(F_\theta(\mathbf{x}), \mathbf{y})$, where the label $\mathbf{y}$ is often represented by a one-hot vector.

**Adversarial Attacks.** An adversary (*i.e.,* attacker) typically optimizes for an additive transformation $\boldsymbol{\delta}$ to the image, $\mathbf{x}_a = \mathbf{x} + \boldsymbol{\delta}$, which can fool the model $F_\theta$ to make incorrect predictions:

$$\mathbf{x}_a = \operatorname*{argmax}_{\mathbf{x}_a} \mathcal{L}(\mathbf{x}_a, \mathbf{y}), \quad \text{s.t.} \quad ||\mathbf{x}_a - \mathbf{x}||_q \leq \epsilon. \tag{1}$$

Here, the magnitude of the added pattern is bounded by a $q$-norm ball of radius $\epsilon$, making the attack perceptually invisible. The role of a defender is to find ways to correct the influence of the attacks and ensure that the model is robust to the adversarial perturbations.

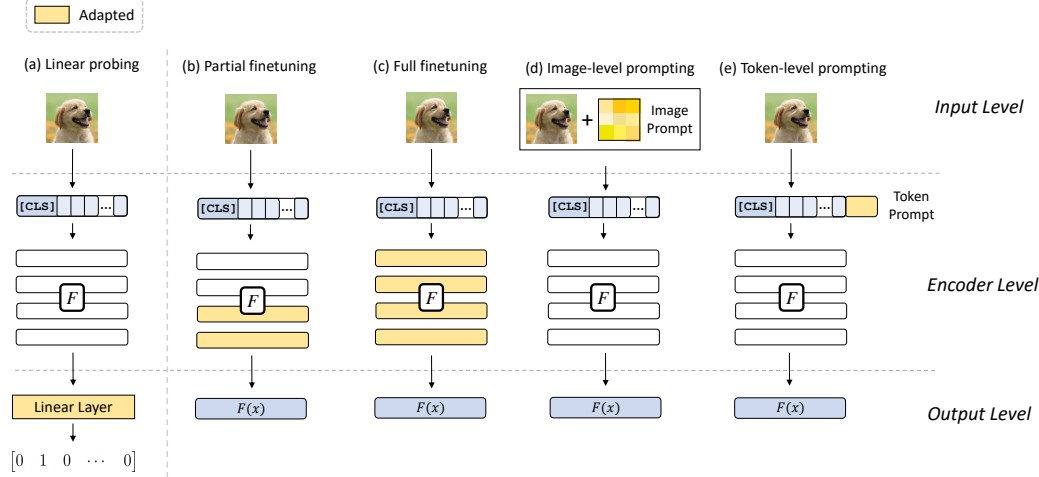

Figure 2: Adaptation methods for large-scale pretrained CLIP model for zero-shot adversarial robustness. (a) Linear probes that adapt the readout layer. (b) Partial finetuning that only updates the last few layers of the model. (c) Finetuning the whole model. (d) Adding visual prompting to the input image. (e) Appending visual prompt tokens to the input token sequence.

**Zero-Shot Adversarial Robustness.** Large-scale vision-language models generalize well to new tasks and datasets at test time. However, the *zero-shot* transferability of adversarial robustness of these models is less explored. In our zero-shot adversarial robustness setup, we study the worst case: we assume that the attacker has the significant advantage of unrestricted access to the ground truth of new tasks at test time, while the defender has no access. While the attacker can directly optimize for an attacked image that fools the model, the defender needs to be robust on all kinds of unseen data and tasks. Compared to the commonly-used robustness setting, which only evaluates robustness on the trained task, our setup is more challenging.

**Adversarial Training** is the common strategy to improve a model's adversarial robustness. By retraining the model on mined adversarial examples, the model is incentivized to learn features invariant under adversarial transformations. The defender often optimizes for the objective

$$\theta = \operatorname*{argmin}_{\theta} \mathcal{L}(F_\theta(\mathbf{x}_a), \mathbf{y}), \tag{2}$$

so that the model $F_\theta$ still makes correct predictions on the generated adversarial examples. As shown in Figure 1, the vanilla adversarial training approach is effective on seen tasks but fails when the model is evaluated on attacked data from unseen tasks.

## 3.2 ADAPTING THE LARGE-SCALE MODELS

One of the key advantages of large-scale pretrained models is that the features of these models are generalizable to various tasks without full retraining. Thus, we would like to make a lightweight adjustment to the models with a small set of training data of attacked images on known tasks, and maximally retain the zero-shot generalization ability of these large models while improving their adversarial robustness on new tasks. We adopt CLIP, one of the best performing vision-language models for zero-shot recognition, as our base model, and investigate a few adaptation strategies as shown in Figure 2 to improve the zero-shot adversarial robustness of CLIP.

**Finetuning (FT).** A typical way to adapt a pretrained model, finetuning, is to update the parameters of the model either entirely or partially. While this approach improves model robustness on the target distribution, Radford et al. (2021) and Pham et al. (2021) show that this improvement often comes at a cost of generalization; directly modifying model parameters may lead to a higher tendency towards overfitting.

**Visual Prompting (VP).** Recently, visual prompt tuning has emerged as a lightweight approach for adapting pretrained large-scale models. Originating from the natural language processing community, the key idea of prompt tuning is to identify prompts to decorate the inputs that effectively query the pretrained models. The computer vision community recently adopted this idea and is starting to explore approaches that can modify input images in pixel space to better adapt large-scale vision

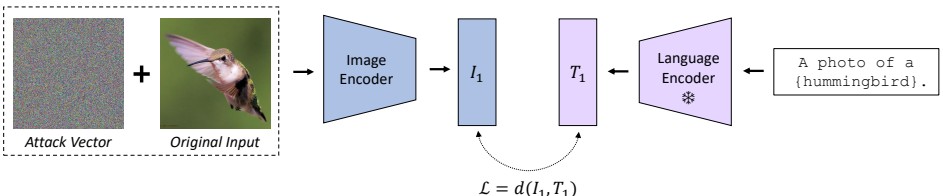

Figure 3: Text-Guided Contrative Adversarial Learning. Instead of using one-hot embedding based supervision, we use adverasrial contrastive learning with language supervision, which achieves better zero-shot robustness transferability.

models for downstream tasks. For transformer-based models, visual prompting methods either learn a token that is appended to the input token sequences (Bar et al., 2022) or learn a direct modification to the input images (Bahng et al., 2022) for adaptation, as shown by (d,e) in Figure 2.

In our study, we conduct an extensive analysis over both adaptation methods to better understand adapting large-scale models for zero-shot adversarial robustness.

### 3.3 Text-Guided Adversarial Contrastive Adversarial Training

Through our initial experiments, we conjecture that the training objective might play a key role in improving the zero-shot adversarial robustness of the models. Radford et al. (2021) indicates that the zero-shot generalization ability of large-scale vision-language models may come from their language supervision. For example, CLIP learns a joint visual-and-text feature space, which helps zero-shot generalization at test time. If we simply finetune the visual encoder with one-hot labels, it may break the joint feature space and harm this zero-shot generalization ability. These observations motivate us to consider using the text information when generating the adversarial examples and also in the training objective during model adaptation.

We now introduce the Text-guided Contrastive Adversarial (TeCoA) training loss, to effectively incorporate text information. In contrast to prior contrastive learning which does image-to-image contrastive learning (Jiang et al., 2020; Kim et al., 2020), we consider a cross-modal text-to-image contrastive learning paradigm. We first generate a set of adversarial examples conditioned on the text inputs which are targeted to fool the model about the correspondence between image features and text embeddings. TeCoA then tries to minimize the feature distance between the attacked image and the correct corresponding text inputs contrastively (Figure 3). We provide additional discussion of this text-image contrastive objective in Appendix A.5.

Concretely, given a set of image and text pairs $\{(\mathbf{x}_i, \mathbf{t}_i)\}$, we use the pretrained CLIP model to encode each pair with an image encoder $F_\theta$ and a text encoder $T$. We then have the following image-to-text contrastive loss function,

$$\mathcal{L}_s(\mathbf{x}, \mathbf{t}, \mathbf{y}) = -\mathbb{E}_{i,j} \left[ \mathbf{y}_{ij} \log \frac{\exp(\cos(\mathbf{z}_i^{(I)}, \mathbf{z}_j^{(T)})/\tau)}{\sum_k \exp(\cos(\mathbf{z}_i^{(I)}, \mathbf{z}_k^{(T)})/\tau)} \right], \tag{3}$$

where the $\mathbf{z}_i^{(I)} = F_\theta(\mathbf{x}_i)$ are the features of the input image, and $\mathbf{z}_i^{(T)} = T(\mathbf{t}_i)$ are the features from the input text. We use $\mathbf{y}_{ij}$ to indicate which image-text pairs are positive and which are negative; this indicator satisfies $\mathbf{y}_{ij} = 1$ if and only if the examples $i = j$, and 0 otherwise. $\tau$ is a scalar hyper-parameter, and $\cos$ denotes the cosine similarity function.

**Constructing Adversarial Examples for Image-Text Pair.** Instead of maximizing the standard cross-entropy loss, we maximize the image-text contrastive loss given a batch of natural images $\mathbf{x}$ and text $\mathbf{t}$:

$$\mathbf{x}_a = \underset{\mathbf{x}_a}{\arg\max} \, \mathcal{L}_s(\mathbf{x}_a, \mathbf{t}, \mathbf{y}), \quad \text{s.t.} \quad ||\mathbf{x}_a - \mathbf{x}||_q < \epsilon. \tag{4}$$

Here, for image $\mathbf{x}_i$ in the batch, the associated text $\mathbf{t}_i$ can be its natural label text or constructed via the standard prompts for the zero-shot tasks (*e.g.*, "a photo of a [LABEL]"). The indicator $\mathbf{y}_{ij} = 1$ when image $\mathbf{x}_i$ has category $j$ as its ground-truth class, and 0 otherwise. In practice, we find this objective is effective at generating adversarial examples for zero-shot image recognition tasks.

**Text-Guided Contrastive Adversarial Training.** Once we have the adversarial examples, we optimize the parameters $\theta$ of the vision encoder $F_\theta$ to minimize the aforementioned objective (Equa-

tion 8) on the generated adversarial examples,

$$\theta = \underset{\theta}{\arg\min}\, \mathcal{L}_s(\mathbf{x}_a, \mathbf{t}, \mathbf{y}). \tag{5}$$

Our algorithm iteratively alternates between generating adversarial examples and updating the model via Equation 5. Since TeCoA uses additional information from the text embeddings to correct the visual features corrupted by the adversarial attacks, it helps the model to retain zero-shot transferability regarding adversarial robustness.

**Contrastive Adversarial Training without Text.** To validate whether the gain of zero-shot robustness comes from language supervision or is due to the formulation of the contrastive loss itself, we consider two variants of contrastive losses which do not utilize languages in our experiments. One is a contrastive adversarial training loss with one-hot labels (`CoAdv.`), where the label information is encoded as a one-hot vector and is used to contrast with the image features. Another is based on Jiang et al. (2020), where the model is finetuned on adversarial examples to fool the image-to-image contrastive loss, denoted as `ImgCoAdv.` More details are presented in Section 4.

## 4 EXPERIMENTS

We start by describing our experiment setups and present the experimental results of 16 datasets in Section 4.1. We observe that CLIP adapted with TeCoA achieves an average of 31 points improvement across the datasets over the original CLIP model. In Section 4.2, we provide extensive analysis over the design choices involved in our approach and identify that the use of language in TeCoA largely improves the model's zero-shot adversarial robustness.

**Datasets.** We evaluate the zero-shot adversarial robustness conferred by TeCoA trained on ImageNet (Deng et al., 2009a) and report the performance of the models on the ImageNet test set as well as 15 zero-shot test datasets, covering a diverse range of recognition tasks. Specifically, we include CIFAR10, CIFAR100 (Krizhevsky et al., 2009), STL10 (Coates et al., 2011), Caltech101 (Fei-Fei et al., 2004), and Caltech256 (Griffin et al., 2007) for generic classification; OxfordPets (Parkhi et al., 2012), StanfordCars (Krause et al., 2013), Food101 (Bossard et al., 2014), Flowers102 (Nilsback & Zisserman, 2008), and FGVCAircraft (Maji et al., 2013) for fine-grained classification; SUN397 (Xiao et al., 2010) for scene recognition; and DTD (Cimpoi et al., 2014) for texture recognition. Finally, we include three datasets with domain-specialized tasks, PatchCamelyon (PCAM, lymph node tumor detection) (Veeling et al., 2018), HatefulMemes (hatespeech detection) (Kiela et al., 2020), and EuroSAT (satellite image classification) (Helber et al., 2017).

**Baselines.** We consider multiple variants to adapt `CLIP` (Radford et al., 2021). For adaptation methods, we consider `visual prompting (VP)` (Bahng et al., 2022), `linear probing (LP)` and `finetuning (FT)`. Since `LP` involves training a linear readout layer on the target task, it is not zero-shot, but we still evaluate it for reference. For training losses, we consider

(1) vanilla `cross-entropy loss (CE)`;
(2) standard `adversarial training loss (Adv.)` with the cross-entropy loss;
(3) `contrastive adversarial training loss (CoAdv.)`;
(4) `contrastive adversarial training over images (ImgCoAdv.)`;
(5) our `text-guided contrastive adversarial training (TeCoA)`.

In our experiments, we name these variants by `adaptation(loss)`. Detailed formulations and associated training algorithms for each loss can be found in Section A.3.

**Implementation Details.** We use CLIP-B/32 architecture. We optimize the model using an SGD optimizer with momentum 0.9. We train for 10 epochs. For finetuning, we use a learning rate of $1e-5$. For visual prompt tuning, we use a learning rate of 40 and a batch size of 256. For prompt tuning on the entire ImageNet dataset, we use token-level prompt with size 200, while for subsets of ImageNet (1K, 5K, and 50K images), we use token-level prompt with size 5 and a smaller batch size of 64. Unless specified, during adversarial training, we generate $L_{\inf} = 1/255$ bounded attacks using a 2-step PGD attack (Madry et al., 2018) with step size $\alpha = 1/255$. We test the robustness of our model using 100 steps of PGD attack, with step size $\alpha = 1/255$.

Table 1: Zero-shot robust accuracies on images attacked with 100 steps of PGD. We show the test accuracy for 16 tasks (columns) and 12 different methods (rows). We show the average performance of each method in the last column. The best accuracies are **bolded**. Standard adversarial finetuning achieves the highest robust accuracy on ImageNet, but this gain comes at the cost of losing robustness on other zero-shot tasks. Adversarial visual prompting improves the average robustness from 10.6 to 31.8. Adding TeCoA with language supervision improves the robustness for both visual prompting and finetuning, where finetuning with TeCoA achieves the best overall robustness, outperforming vanilla CLIP by 31 points.

| | **DATASETS** | | | | | | | | | | | | | | | | |
| Method | ImageNet | CIFAR10 | STL-10 | CIFAR100 | SUN | StanfordCars | Food101 | OxfordPet | Flower102 | DTD | EUROSAT | FGVC | HateMemes | PCAM | Caltech101 | Caltech256 | Average |
|---|---|---|---|---|---|---|---|---|---|---|---|---|---|---|---|---|---|
| CLIP | 1.72 | 9.57 | 35.40 | 4.55 | 1.02 | 0.27 | 3.95 | 2.72 | 1.19 | 2.5 | 0.04 | 0.00 | 2.20 | 0.10 | 24.63 | 15.27 | 6.57 |
| +VPT (CE) | 2.27 | 13.25 | 40.74 | 5.59 | 1.60 | 0.21 | 4.21 | 3.24 | 1.77 | 3.56 | 0.01 | 0.00 | 2.47 | 0.21 | 28.46 | 17.17 | 7.79 |
| +LP (CE) | 5.78 | 22.07 | 36.52 | 16.02 | 12.11 | 0.20 | 5.625 | 0.00 | 0.00 | 0.00 | 6.06 | 0.00 | 44.53 | 50.11 | 36.72 | 20.12 | 16.00 |
| +FT (CE) | 9.13 | 27.60 | 61.33 | 8.50 | 4.74 | 1.12 | 10.13 | 25.81 | 5.58 | 2.61 | 0.52 | 0.00 | 1.07 | 1.15 | 44.98 | 35.44 | 14.98 |
| +VPT (Adv.) | 17.50 | 45.39 | 72.52 | 22.10 | 18.15 | 9.65 | 16.02 | 44.53 | 25.06 | 20.27 | 2.12 | 3.39 | **54.30** | **52.57** | 61.93 | 44.06 | 31.84 |
| +VPT (CoAdv.) | 17.20 | 40.72 | 72.11 | 20.53 | 18.01 | 10.26 | 15.04 | 42.71 | 22.15 | 21.65 | 2.80 | 2.79 | 26.57 | 13.69 | 62.03 | 43.23 | 26.97 |
| +VPT (ImgCoAdv.) | 1.07 | 8.35 | 17.44 | 2.12 | 2.58 | 0.03 | 0.94 | 3.54 | 1.94 | 7.39 | 0.10 | 0.03 | 6.37 | 14.96 | 14.38 | 6.91 | 5.51 |
| VPT (TeCoA) | 23.71 | 49.18 | 75.54 | 25.82 | 24.87 | 10.91 | 20.10 | 48.57 | 23.73 | 21.11 | 11.26 | 4.05 | 31.13 | 30.50 | 63.79 | 47.66 | 32.00 |
| +FT(Adv.) | **45.12** | 11.04 | 25.90 | 1.51 | 0.52 | 0.25 | 0.56 | 20.40 | 0.11 | 2.77 | 3.62 | 0.48 | 23.12 | 24.86 | 6.68 | 2.92 | 10.62 |
| +FT (CoAdv.) | 33.41 | 2.51 | 5.92 | 0.40 | 0.01 | 0.14 | 0.09 | 0.19 | 0.47 | 0.21 | 6.62 | 0.42 | 30.2 | 44.7 | 0.06 | 0.08 | 7.83 |
| +FT (ImgCoAdv.) | 0.03 | 4.00 | 3.76 | 0.37 | 0.12 | 0.05 | 0.17 | 0.57 | 0.21 | 0.96 | 4.13 | 0.18 | 32.93 | 26.41 | 0.33 | 0.18 | 4.65 |
| FT (TeCoA) | 41.88 | **59.28** | **83.45** | **34.13** | **29.81** | **13.37** | **27.99** | **62.61** | **30.69** | **22.88** | **15.18** | **5.10** | 31.97 | 23.87 | **69.07** | **59.54** | **38.18** |

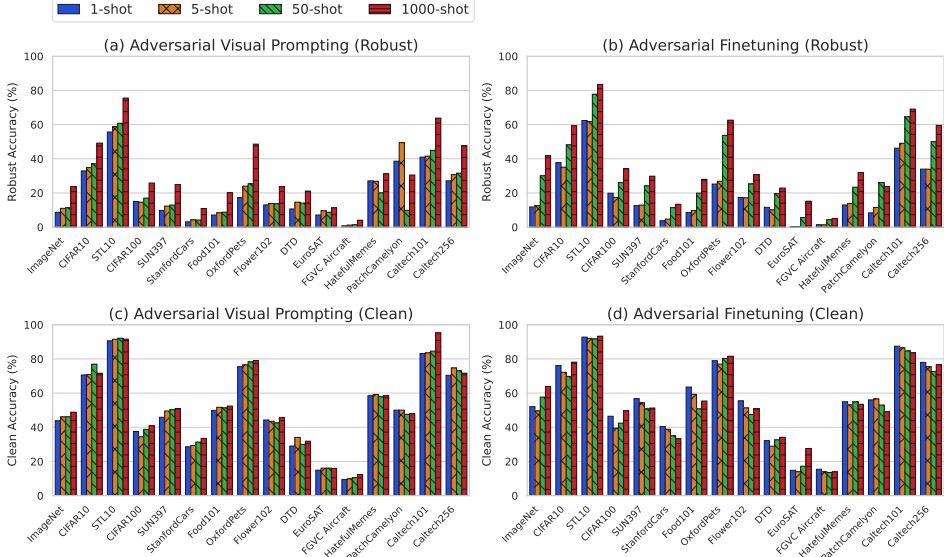

Figure 4: Effect of training set size. We consider several setups where only a restricted number of samples from each training class are available during adversarial training with TeCoA. Training on more data in general improves zero-shot robustness, but not affect clean performance much.

## 4.1 EXPERIMENTAL RESULTS

We first use PGD attack with 100 steps to evaluate the robustness of CLIP models that are adapted on ImageNet. In Table 1, we show the robust accuracies for the models on the Imagenet test set and 15 zero-shot recognition tasks (*i.e.,* the training classes and test classes are non-overlapping). Each row is the robust accuracy for one method, where we compare the proposed TeCoA with VP and FT against 10 baselines and their variants. We **bold** the best results for each dataset (column).

From Table 1, we can see that most adapted models with adversarial training losses except for `ImgCoAdv.` have better robust accuracies compared to the standard cross-entropy loss. If using adversarial training, the vanilla `FT (Adv.)`, which finetunes the entire model with adversarial training, achieves the best adversarial robustness on ImageNet (non zero-shot data, same as the training tasks) while the average robust accuracy is 10.62%, which is only slightly better than the original CLIP (6.57%). In the meantime, `VP(Adv.)` achieves much better results, improving the

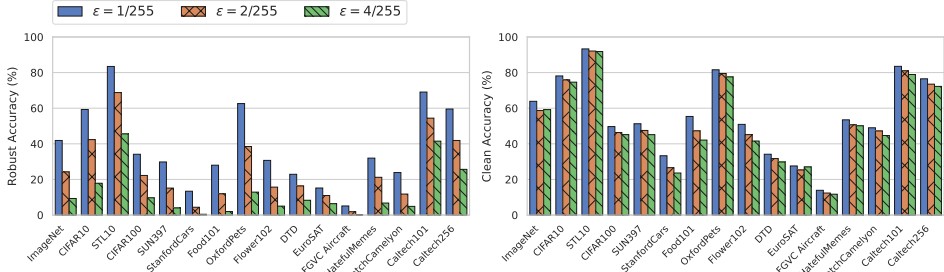

Figure 5: Zero-shot adversarial robustness under different perturbation bounds ($\epsilon = 1, 2, 4/255$). We vary the perturbation bound for adversarial finetuning with TeCoA. Each adapted model is evaluated under attacks from the same bound seen during training. We show both the robust accuracy (left) and clean accuracy (right). Our defense is still effective on zero-shot tasks when the perturbation gets larger.

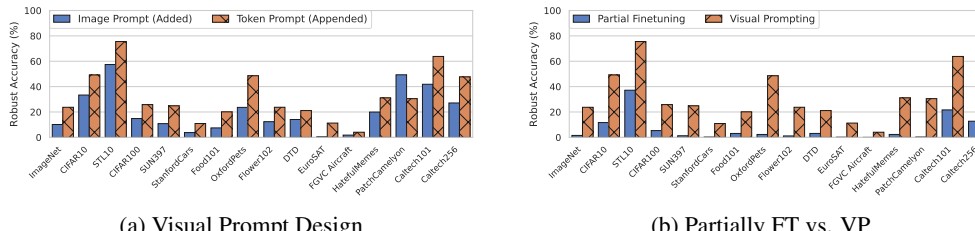

(a) Visual Prompt Design                (b) Partially FT vs. VP

Figure 6: (a, left) We conduct an ablation study of whether to add visual prompt to the input image or to append prompt token to the input sequence. (b, right) We optimize the same amount of parameters in partial finetuning and VP, we find VP is more effective when only a small number of parameters are optimized.

accuracy number from 6.57% to 31.84%. This indicates that visual prompting is more powerful than finetuning when coupled with the standard adversarial training.

Within each set of the variants using the same adaptation method, we compare the effectiveness of different training losses. We notice that both `CoAdv.` and `ImgCoAdv.` are much worse than the proposed `TeCoA` and even the standard adversarial training (`Adv.`). This indicates that the formulation of contrastive learning may not necessarily help improve the zero-shot adversarial robustness and the use of the language supervision might.

Overall, we find that adapting CLIP with TeCoA using model finetuning presents the best performance across the datasets, improving the accuracy number from 6.57% to 38.18%, roughly 31 points. This might look counter-intuitive as VP significantly outperforms FT under the standard adversarial training without texts. One hypothesis is that with sufficient semantic information at test time, finetuning that directly modifies the model parameters may be more expressive than just modifying the inputs given more model parameters are tuned. In Table 3 in Appendix, we show the clean accuracy of the same models, which gives similar trend as the robust accuracies. We also show results under $\epsilon = 1/255$ and $\epsilon = 4/255$ using AutoAttack Croce & Hein (2020) in Table 4 and Table 5, where our model is still more robust than baselines, up to an average of 36 points. We describe details in Section A.2.

## 4.2 ANALYSIS

**Training Set Size** is an important factor when adapting large-scale models. In Figure 4, we show the results of adapting CLIP with TeCoA with 1, 5, 50, and 1000 shot per category on ImageNet. We observe that increasing the amount of training data improves robust accuracy. Moreover, we also find that `FT(TeCoA)` outperforms the non-adapted CLIP even when the model is adapted with just one shot per class (the blue bars in Figure 4).

**Effect of Attack Strength.** To validate whether our method works when the adversarial perturbations become larger, we increase the perturbation bound for our TeCoA adversarial training. Figure 5 shows that, while increasing the attack strength decreases the robust accuracy, our model can still transfer robustness to zero-shot tasks.

**Visual Prompt Designs.** There are two ways to design visual prompts. One is to append additional tokens to image tokens and the other is to add small decoration (*i.e.,* learnable noises) to the in-

Figure 7: Balancing model robustness and clean accuracy via weight interpolation. By changing the interpolation ratio for the adapted CLIP and vanilla CLIP weights, we can dynamically improve clean accuracy or robust accuracy. There is a win-win sweet spot, indicated by the star, where we improves on both zero-shot robust accuracy and zero-shot clean accuracy.

Table 2: Are labels required to adapt large-scale models for zero-shot adversarial robustness? We find we can achieve similar zero-shot adversarial robustness by adapting the model on unlabeled data using our algorithm.

| | | DATASETS | | | | | | | | | | | | | | | | |
|---|---|---|---|---|---|---|---|---|---|---|---|---|---|---|---|---|---|---|
| Method | Labeled? | ImageNet | CIFAR10 | STL-10 | CIFAR100 | SUN | StanfordCars | Food101 | OxfordPet | Flower102 | DTD | EUROSAT | FGVC | HateMemes | PCAM | Caltech101 | Caltech256 | Average |
| VPT (TeCoA) Robust Accuracy | Yes | 23.71 | 49.18 | 75.54 | 25.82 | 24.87 | 10.91 | 20.10 | 48.57 | 23.73 | 21.11 | 11.26 | 4.05 | 31.13 | 30.50 | 63.79 | 47.66 | 32.00 |
| VPT (TeCoA) Robust Accuracy | No | 23.79 | 48.73 | 75.43 | 24.83 | 22.58 | 12.05 | 20.38 | 47.26 | 24.54 | 21.22 | 12.49 | 3.48 | 26.40 | 40.22 | 65.55 | 49.41 | **32.40** |
| VPT (TeCoA) Clean Accuracy | Yes | 48.84 | 71.63 | 91.59 | 40.96 | 50.97 | 33.50 | 52.41 | 79.07 | 45.78 | 31.86 | 15.93 | 12.24 | 58.53 | 47.85 | 95.19 | 71.66 | 53.00 |
| VPT (TeCoA) Clean Accuracy | No | 59.11 | 70.65 | 91.31 | 39.41 | 47.27 | 37.87 | 55.39 | 78.25 | 46.17 | 32.44 | 19.47 | 12.69 | 55.83 | 49.60 | 86.34 | 73.60 | **53.46** |
| FT (TeCoA) Robust Accuracy | Yes | 41.88 | 59.28 | 83.45 | 34.13 | 29.81 | 13.37 | 27.99 | 62.61 | 30.69 | 22.88 | 15.18 | 5.10 | 31.97 | 23.87 | 69.07 | 59.54 | **38.18** |
| FT (TeCoA) Robust Accuracy | No | 38.76 | 60.90 | 79.41 | 31.56 | 30.33 | 13.87 | 27.72 | 58.84 | 29.48 | 22.50 | 11.67 | 5.10 | 29.97 | 19.14 | 67.49 | 59.41 | 36.63 |
| FT (TeCoA) Clean Accuracy | Yes | 63.87 | 78.12 | 93.30 | 49.68 | 51.28 | 33.30 | 55.37 | 81.58 | 50.92 | 34.15 | 27.57 | 13.89 | 53.47 | 49.01 | 83.51 | 76.51 | 55.97 |
| FT (TeCoA) Clean Accuracy | No | 58.87 | 80.51 | 91.18 | 49.08 | 54.93 | 39.39 | 59.14 | 77.62 | 50.79 | 34.20 | 26.13 | 14.52 | 52.10 | 52.97 | 81.39 | 77.93 | **56.30** |

put images. From Figure 6a, we can see appending learnable tokens in the input token sequences achieves consistently better performance than just adding the prompt value to the images.

**Number of Adapted Parameters.** The number of parameters that are adapted during training highly affects model performance. VP is light-weight, as it only modifies the inputs, while FT may either adjust part of the model or the entire model. In Figure 6b, we show the comparison of partially finetuning and visual prompting of CLIP. We can see that with the same amount of parameters adapted, visual prompting is more effective in adapting the large scale model for zero-shot adversarial robustness.

**Are Labels Required for Adapting CLIP?** Unlabeled data is rich. We investigate if it's necessary to use the groundtruth labels from images. We generate the pseudo text labels that CLIP retrieves from clean images, where we show details in Section A.3.5. We then run our TeCoA on the pseudo labels in the same way as the labelled data. We show experimental results in Table 2, where we obtain a similar zero-shot robust accuracy even without labels.

**Trading off between Robust Accuracy and Clean Accuracy.** Similar to typical adversarial training, TeCoA also poses a trade-off between the clean accuracy and the robust accuracy Tsipras et al. (2019). Ideally, we want to be able to dynamically adjust this trade-off depending on the desired level of adversarial robustness. Here we can balance this trade-off by using model weights interpolation (Wortsman et al., 2022). In Figure 7, we can see there is a sweet spot in interpolating the model where we improve both the robustness and clean accuracy, marked by the star in the figure.

## 5    CONCLUSION

In this paper, we provided a holistic study of the zero-shot adversarial robustness problem of large-scale vision-language models. We identified the effects of various adaption methods and training losses when adapting models, and conjectured that existing methods failed to generalize to new tasks due to the lack of the language supervision. We proposed a text-guided contrastive adversarial training (TeCoA) which can be used with model finetuning and visual prompting to drastically improve the zero-shot adversarial robustness of CLIP. Extensive experimental evaluation showed the effectiveness of TeCoA, and the detailed analyses provide useful lessons for adapting large-scale models to improve their zero-shot adversarial robustness, shedding light on this important problem.

## 6 ACKNOWLEDGEMENT

This research is based on work partially supported by the DARPA SAIL-ON program, the DARPA MCS program, the NSF NRI Award #2132519, a GE/DARPA grant, a CAIT grant, and gifts from JP Morgan, DiDi, and Accenture.

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

Table 3: Zero-shot clean accuracy. After adapting the CLIP model on ImageNet, the zero-shot performance on clean image in general drops. TeCoA based methods achieves the relatively small drop compared with other training object when the model has been adversarially trained.

| | DATASETS | | | | | | | | | | | | | | | | |
|---|---|---|---|---|---|---|---|---|---|---|---|---|---|---|---|---|---|
| Method | ImageNet | CIFAR10 | STL-10 | CIFAR100 | SUN | StanfordCars | Food101 | OxfordPet | Flower102 | DTD | EUROSAT | FGVC | HatefulMemes | PCAM | Caltech101 | Caltech256 | Average |
| CLIP | 59.12 | 89.06 | 97.16 | 62.32 | 57.67 | 52.14 | 83.15 | 85.82 | 65.64 | 40.52 | 38.26 | 20.04 | 56.20 | 52.03 | 91.75 | 82.04 | 64.56 |
| +VP (CE) | 56.45 | 86.72 | 95.90 | 59.35 | 59.08 | 47.61 | 76.74 | 85.91 | 60.48 | 12.81 | 33.85 | 19.05 | 54.90 | 50.42 | 89.93 | 79.19 | 60.52 |
| +LP (CE) | 61.88 | 91.80 | 97.66 | 58.01 | 63.44 | 9.77 | 60.39 | 39.84 | 3.71 | 3.91 | 49.41 | 7.42 | 44.53 | 52.29 | 67.19 | 66.02 | 48.58 |
| +FT (CE) | **72.72** | 87.02 | 97.25 | 63.93 | 59.90 | 47.69 | 77.96 | 87.19 | 63.47 | 40.80 | 37.22 | 16.29 | 57.00 | 43.90 | 87.38 | 82.75 | 63.90 |
| +VP (TeCoA) | 10.13 | 33.36 | 57.42 | 14.84 | 10.82 | 3.78 | 7.45 | 23.63 | 12.36 | 14.09 | 0.35 | 1.77 | 19.93 | 49.33 | 41.85 | 27.11 | 20.51 |
| +VP (Adv.) | 42.34 | 69.06 | 90.63 | 37.52 | 44.31 | 35.51 | 50.59 | 77.60 | 49.60 | 34.10 | 14.97 | 10.92 | 54.30 | 52.47 | 84.18 | 69.68 | 51.11 |
| +VP (CoAdv.) | 42.05 | 64.16 | 25.57 | 90.61 | 43.72 | 35.94 | 49.43 | 77.13 | 47.65 | 35.05 | 12.41 | 10.98 | 57.33 | 45.54 | 84.37 | 69.25 | 49.45 |
| +VP (ImgCoAdv.) | 13.24 | 27.74 | 56.07 | 8.80 | 20.37 | 4.07 | 18.79 | 34.75 | 17.32 | 23.78 | 6.54 | 4.26 | 48.37 | 55.55 | 52.05 | 32.03 | 26.48 |
| +VP (TeCoA) | 48.84 | 71.63 | 91.59 | 40.96 | 50.97 | 33.50 | 52.41 | 79.07 | 45.78 | 31.86 | 15.93 | 12.24 | 58.53 | 47.85 | 95.19 | 71.66 | 53.00 |
| +FT (Adv.) | 61.55 | 26.01 | 41.82 | 5.52 | 1.51 | 0.38 | 2.31 | 27.72 | 0.98 | 6.22 | 12.21 | 1.38 | 47.83 | 41.62 | 12.75 | 6.09 | 18.49 |
| +FT (CoAdv.) | 50.18 | 10.52 | 17.61 | 0.63 | 0.18 | 0.60 | 0.90 | 2.18 | 1.31 | 1.22 | 19.47 | 1.32 | 53.20 | 51.72 | 0.57 | 0.44 | 13.25 |
| +FT (ImgCoAdv.) | 0.13 | 10.44 | 11.20 | 1.03 | 0.27 | 0.12 | 1.13 | 2.94 | 0.91 | 2.02 | 13.57 | 1.11 | 50.60 | 37.97 | 1.13 | 0.55 | 8.45 |
| +FT (TeCoA) | 63.87 | 78.12 | 93.30 | 49.68 | 51.28 | 33.30 | 55.37 | 81.58 | 50.92 | 34.15 | 27.57 | 13.89 | 53.47 | 49.01 | 83.51 | 76.51 | 55.97 |

Kaiyang Zhou, Jingkang Yang, Chen Change Loy, and Ziwei Liu. Learning to prompt for vision-language models. *International Journal of Computer Vision*, 130(9):2337–2348, 2022c.

# A APPENDIX

## A.1 EXPERIMENTS

### A.1.1 ZERO-SHOT CLEAN ACCURACY OF OUR ADAPTED MODEL

We show the results for accuracy on clean images in Table 3.

## A.2 AUTOATTACK EXPERIMENT

We also consider a stronger attack AutoAttack Croce & Hein (2020) in our evaluation. Since our method uses adversarial training and does not rely on the obfuscated gradient, we use two APGD variants, APGD-CE and APGD-DLR, in AutoAttack to evaluate. We show robust accuracy under perturbation bound $\epsilon = 1/255$ in Table 4 and robust accuracy under perturbation bound $\epsilon = 4/255$ in Table 5. For both perturbation bounds, our method achieves higher robust accuracy than vanilla CLIP by up to 36 points on average.

Notably, on $\epsilon = 1/255$, *even evaluated under the stronger AutoAttack, the robustness accuracy of FT (TeCoA) is 37.02, which is still higher than all other baselines methods from Table 1 in robust accuracy, even though those baselines are evaluated under a weaker PGD100 attack.* A larger perturbation bound $\epsilon = 4/255$ makes the attack stronger, where our method still improves robustness by an average of 9 points. In addition, while the AutoAttack significantly reduces the robust accuracy of CLIP from 6.57 to 0.53, it only slightly decreases our TeCoA's robust accuracy: 2.1 points for visual prompt tuning and 1.16 for finetuning model (see Table 1 and Table 4).

One reason for AutoAttack to be so effective in attacking vanilla CLIP than PGD100 is because it uses a fractional attack vector, which is not rounded by 1/255 during the inference. Images are often encoded via integer from 0 to 255, which allows only attack at the integer level. In the main paper, we use PGD attacks with step size 1 (if the image ranges from 0 to 1, then step size is proportionally 1/255) for 100 steps. Since there is no fractional attack value, the attack space is constrained and less effective. This is because the standard image inputs have value resolutions that should be larger than 0.5, and any values smaller than this would be rounded when encoding the images. If we ignore the fact that images will be encoded in integers between 0 to 255, then we can have stronger attacks by exploring the fraction values. Since the AutoAttack automatically reduces the attack step size when loss oscillates, it explores the fraction space and is more effective in the attack.

Table 4: Zero-shot robust accuracies on images attacked with AutoAttack with $\epsilon = 1/255$. We use the CE and DLR variants of APGD in AutoAttack, which is a stronger attacker, where the adaptive step size can be a fraction (for PGD100, the minimum step size is 1). We show the test accuracy for 16 tasks (columns) and 12 different methods (rows). We show the average performance of each method in the last column. The best accuracies are **bolded**. Autoattack is a stronger attack than PGD, which significantly reduces the robustness accuracy of vanilla CLIP. For example, CLIP robust accuracy on Caltech101 was reduced from 24.63 (PGD100) to 0.41 (AA). However, AA only slightly decreases our TeCoA's robustness, from 69.07 to 68.40. By our language-guided adversarial training (TeCoA), we improve the robustness by an average of 36 points than CLIP, which is a larger robustness gain than the PGD100 experiment. In addition, compared with PGD 100 attack, the average robustness only decreases by around 1 point on FT (TeCoA), suggesting that our TeCoA is robust and stronger attack cannot further decrease the accuracy.

| | **DATASETS** | | | | | | | | | | | | | | | | |
| Method | ImageNet | CIFAR10 | STL-10 | CIFAR100 | SUN | StanfordCars | Food101 | OxfordPet | Flower102 | DTD | EUROSAT | FGVC | HateMemes | PCAM | Caltech101 | Caltech256 | Average |
|---|---|---|---|---|---|---|---|---|---|---|---|---|---|---|---|---|---|
| CLIP | 0.00 | 2.73 | 2.96 | 1.21 | 0.04 | 0.11 | 0.02 | 0.03 | 0.07 | 0.11 | 0.11 | 0.18 | 0.10 | 0.16 | 0.41 | 0.15 | 0.53 |
| VPT (TeCoA) | 20.82 | 47.25 | 74.99 | 23.39 | 19.76 | 8.63 | 16.81 | 44.78 | 20.90 | 18.83 | 10.73 | 3.24 | 31.13 | 29.35 | 62.19 | 45.64 | 29.90 |
| FT (TeCoA) | **39.81** | **58.27** | **83.16** | **32.57** | **29.03** | **12.03** | **25.79** | **61.76** | **28.93** | **20.70** | **13.26** | **4.05** | **31.83** | **24.09** | **68.40** | **58.59** | **37.02** |

Table 5: Zero-shot robust accuracies on images attacked with AutoAttack with $\epsilon = 4/255$. We show the test accuracy for 16 tasks (columns) and 12 different methods (rows). We show the average performance of each method in the last column. The best accuracies are **bolded**. Autoattack is a stronger attack than PGD, which reduces the robustness accuracy of CLIP even more. By our language-guided adversarial training (TeCoA), we improve the robustness by an average of 9 points to CLIP.

| | **DATASETS** | | | | | | | | | | | | | | | | |
| Method | ImageNet | CIFAR10 | STL-10 | CIFAR100 | SUN | StanfordCars | Food101 | OxfordPet | Flower102 | DTD | EUROSAT | FGVC | HateMemes | PCAM | Caltech101 | Caltech256 | Average |
|---|---|---|---|---|---|---|---|---|---|---|---|---|---|---|---|---|---|
| CLIP | 0.05 | 0.01 | 0.00 | 0.04 | 0.04 | 0.15 | 0.02 | 0.00 | 0.05 | 0.00 | 0.11 | **0.24** | 0.06 | 0.18 | 0.00 | 0.04 | 0.06 |
| VPT (TeCoA) | 1.44 | 8.99 | 21.83 | 3.78 | 0.78 | 0.25 | 0.46 | 2.43 | 2.80 | 4.41 | 10.04 | 0.21 | 19.73 | 27.74 | 11.79 | 5.66 | 7.65 |
| FT (TeCoA) | **6.40** | **15.36** | **34.30** | **8.33** | **2.94** | **0.35** | **1.12** | **7.88** | **4.16** | **6.44** | **4.43** | 0.18 | **5.63** | **3.17** | **26.85** | **17.09** | **9.04** |

## A.3 TRAINING LOSSES AND ALGORITHMS

We give formulations for the different training algorithms considered in our experiments. Throughout this section, let $F_\theta$ denote an image encoder parameterized by $\theta$, and let $T$ denote a frozen text encoder. Let $\mathbb{D}$ denote a dataset containing pairs $(x, y)$ of images and their respective one-hot labels.

### A.3.1 STANDARD ADVERSARIAL TRAINING WITH CROSS-ENTROPY LOSS (ADV.)

The standard adversarial training paradigm. We initialize a learnable linear layer $C_\phi$, and append it to $F_\theta$. The classification loss is $\mathcal{L}(C_\phi(F_\theta(\mathbf{x}_a)), \mathbf{y})$ defined in cross-entropy loss.

We first train $C_\phi$ on standard images. Then, given a natural image $\mathbf{x}$ with one-hot label $\mathbf{y}$, we generate an attacked image $\mathbf{x}_a$ by maximizing the loss $\mathcal{L}(C_\phi(F_\theta(\mathbf{x}_a)), \mathbf{y})$. We then update $\theta$ to minimize the loss $\mathcal{L}(C_\phi(F_\theta(\mathbf{x}_a)), \mathbf{y})$. We describe our algorithm in Algorithm 1.

---

**Algorithm 1** Standard Adversarial Training (Adv.)

---

**Input:** Dataset $\mathbb{D}$, learnable parameter $\theta$, model $F$, parameter of projector $C_\phi$
    **for all** iter $\in$ preset number of training epochs **do**
        **for all** $\mathbf{x}, \mathbf{y} \in$ minibatch **do**
            $\mathbf{x}^a = \mathrm{argmax}_{\mathbf{x}_a} \mathcal{L}(C_\phi(F_\theta(\mathbf{x}^a)), \mathbf{y})$         ▷ Generating adversarial attacks
            $\theta = \theta - \nabla_\theta \mathcal{L}(C_\phi(F_\theta(\mathbf{x}_a)), \mathbf{y})$     ▷ Training on generated adversarial examples
        **end for**
    **end for**

---

### A.3.2 Contrastive Adversarial Training Loss (CoAdv.)

We study how much the contrastive learning objective contributes to the zero-shot robustness gain. Instead of using one-hot label $y$ and cross-entropy loss in our objective, we create a dictionary of embeddings $\mathcal{E}$ by random initialization, where each embedding $\mathbf{e}_i$ denotes the code representation for the category $y_i$. We optimize the following contrastive learning loss:

$$\mathcal{L}_s(\mathbf{x}, \mathcal{E}, \mathbf{y}) = -\mathbb{E}_{i,j} \left[ \mathbf{y}_{ij} \log \frac{\exp(\cos(\mathbf{z}_i^{(I)}, \mathbf{e}_j)/\tau)}{\sum_k \exp(\cos(\mathbf{z}_i^{(I)}, \mathbf{e}_j)/\tau)} \right], \tag{6}$$

where the $\mathbf{z}_i^{(I)} = F_\theta(\mathbf{x}_i)$ are the features of the input image, and $\mathbf{e}_j$ are the code representation from the dictionary. We use $\mathbf{y}_{ij}$ to indicate which image-code pairs are positive and which are negative; this indicator satisfies $\mathbf{y}_{ij} = 1$ if and only if the examples $i = j$, and 0 otherwise. $\tau$ is a scalar hyper-parameter, and $\cos$ denotes the cosine similarity function. We describe our algorithm in Algorithm 2.

---

**Algorithm 2** Contrastive Adversarial Training Loss (CoAdv.)

---

**Input:** Dataset $\mathbb{D}$, learnable parameter $\theta$, model $F$
  **for all** iter $\in$ preset number of training epochs **do**
    **for all** $\mathbf{x}, \mathbf{y} \in$ minibatch **do**
      $\mathbf{x}^a = \operatorname{argmax}_{\mathbf{x}^a} \mathcal{L}_s(\mathbf{x}, \mathcal{E}, \mathbf{y})$       ▷ Generating adversarial attacks for contrastive loss
      $\theta = \theta - \nabla_\theta \mathcal{L}_s(\mathbf{x}, \mathcal{E}, \mathbf{y})$      ▷ Contrastive learning on generated adversarial examples
    **end for**
  **end for**

---

### A.3.3 Contrastive Adversarial Training over Images (ImgCoAdv.)

Prior work Jiang et al. (2020) uses image-only contrastive adversarial learning to obtain robustness. We adapt this method as a baseline to study whether using the knowledge from only images — not language — can achieve zero-shot robustness. For each image $\mathbf{x}_i$, we create a transformationed $\mathbf{x}_j$, and form the image pair $(x_i, x_j)$.

We use the same visual encoder to embed the images $\mathbf{x}_i$ and $\mathbf{x}_j$ to obtain the features $\mathbf{z}_i$ and $\mathbf{z}_j$. We then construct the following contrastive learning loss:

$$\mathcal{L}_s(\mathbf{x}_i, \mathbf{x}_j, \mathbf{y}) = -\mathbb{E}_{i,j} \left[ \mathbf{y}_{ij} \log \frac{\exp(\cos(\mathbf{z}_i, \mathbf{z}_j)/\tau)}{\sum_k \exp(\cos(\mathbf{z}_i, \mathbf{z}_j)/\tau)} \right], \tag{7}$$

where the $\mathbf{z}_i = F_\theta(\mathbf{x}_i)$ are the features of the input image. We use $\mathbf{y}_{ij}$ to indicate which image pairs are positive and which are negative; this indicator satisfies $\mathbf{y}_{ij} = 1$ if and only if the images $\mathbf{x}_i$ and $\mathbf{x}_j$ are augmented from the same instance, and 0 otherwise. $\tau$ is a scalar hyper-parameter, and $\cos$ denotes the cosine similarity function.

Let $\mathbf{z}_i^a = F_\theta(\mathbf{x}_i^a)$, where $\mathbf{x}_i^a$ denotes the generated adversarial examples. Then we can obtain the adversarial examples via:

$$\mathbf{x}_i^a = \operatorname*{argmax}_{\mathbf{x}_i^a} \mathcal{L}_s(\mathbf{x}_i, \mathbf{x}_j, \mathbf{y}) = \operatorname*{argmax}_{\mathbf{x}_i^a} -\mathbb{E}_{i,j} \left[ \mathbf{y}_{ij} \log \frac{\exp(\cos(\mathbf{z}_i^a, \mathbf{z}_j)/\tau)}{\sum_k \exp(\cos(\mathbf{z}_i^a, \mathbf{z}_j)/\tau)} \right], \tag{8}$$

Once we generate the adversarial images, we conduct contrastive learning on adversarial images and the paired clean images using Equation 7.

We introduce our algorithm in Algorithm 3.

### A.3.4 Text-Guided Contrastive Adversarial Training (TeCoA)

We describe the TeCoA training algorithm in Algorithm 4. We denote the learnable parameters to be $\theta$. For the visual prompt tuning, $\theta$ is only the prompt vector. For the finetuning method, $\theta$ is the parameter of the whole model.

---

**Algorithm 3** Contrastive Adversarial Training over Images (ImgCoAdv.)

---

**Input:** Dataset $\mathbb{D}$, learnable parameter $\theta$, model $F$
    **for all** iter $\in$ preset number of training epochs **do**
        **for all** $\mathbf{x} \in$ minibatch **do**
            $\mathbf{x}_i^a = \mathrm{argmax}_{\mathbf{x}_i^a}\,\mathcal{L}_s(\mathbf{x}_i^a, \mathbf{x}_j, \mathbf{y})$        $\triangleright$ Generating adversarial attacks for contrastive loss
            $\theta = \theta - \nabla_\theta \mathcal{L}_s(\mathbf{x}_i^a, \mathbf{x}_j, \mathbf{y})$     $\triangleright$ Contrastive learning on generated adversarial examples
        **end for**
    **end for**

---

---

**Algorithm 4** TeCoA Training

---

**Input:** Dataset $\mathbb{D}$, learnable parameter $\theta$, model $F$, text $\mathbf{t}$
    **for all** iter $\in$ preset number of training epochs **do**
        **for all** $\mathbf{x}, \mathbf{y} \in$ minibatch **do**
            $\mathbf{x}^a = \mathrm{argmax}_{\mathbf{x}}\,\mathcal{L}_s(\mathbf{x}, \mathbf{t}, \mathbf{y})$        $\triangleright$ Generating adversarial attacks for contrastive loss
            $\theta = \theta - \nabla_\theta \mathcal{L}_s(\mathbf{x}^a, \mathbf{t}, \mathbf{y})$     $\triangleright$ Contrastive learning on generated adversarial examples
        **end for**
    **end for**

---

### A.3.5 TeCoA Learning on Unlabeled Data

Given an unlabeled image, we first provide a list of text using the possible category names:

```
A photo of a {Category Name}.
```

Since the unlabeled images are not attacked, CLIP can retrieve the nearest text embedding from the image embedding and use the text as the pseudo label for the image. We then conduct the TeCoA training on the images and their pseudo text label. We describe the algorithm below in Algorithm 5.

---

**Algorithm 5** TeCoA Training on Unlabeled Data

---

**Input:** Dataset $\mathbb{D}$ without label, learnable parameter $\theta$, model $F$, text $\mathbf{t}$.
    **for all** iter $\in$ preset number of training epochs **do**
        **for all** $x \in$ minibatch $B = \{x_1, \ldots, x_m\}$ **do**
            $\mathbf{y} = \mathrm{argmin}_{\mathbf{y}}\,\mathcal{L}_s(\mathbf{x}, \mathbf{t}, \mathbf{y})$      $\triangleright$ Finding pseudo label for the clean images using CLIP
            $\mathbf{x}^a = \mathrm{argmax}_{\mathbf{x}}\,\mathcal{L}_s(\mathbf{x}, \mathbf{t}, \mathbf{y})$        $\triangleright$ Generating adversarial attacks for contrastive loss
            $\theta = \theta - \nabla_\theta \mathcal{L}_s(\mathbf{x}^a, \mathbf{t}, \mathbf{y})$     $\triangleright$ Contrastive learning on generated adversarial examples
        **end for**
    **end for**

---

### A.4 Discussion for Results

In Table 1, our TeCoA performs better than existing methods except for LP(CE) and VPT(Adv.) on HateMemes and PCAM datasets. This is because the HateMemes dataset is a binary classification task for detecting hateful speech, which is a very different domain from the ImageNet classification. Since both LP(CE) and VPT (adv) only adapt a small number of parameters on the ImageNet set, the resulting model may overfit less to the Image recognition task, and just perform random guessing. Note that the 54% accuracy is close to random guessing 50%. In addition, PCAM is a binary classification for lymph nodes (medical image, https://github.com/basveeling/pcam), which is also a very different domain from ImageNet. Similar to HateMemes, adapting fewer parameters makes the model learn less and overfit less, where the 52.5% accuracy is close to random guessing 50%. Thus, both datasets remain a big challenge for all existing zero-shot robust classification tasks.

## A.5 DISCUSSION FOR TECOA LOSS

In the main paper, we interpret our loss through the image-text contrastive objective, which first conducts a matrix multiplication between the image embedding and language embedding, and then applies a cross-entropy loss on the output. Since the text embedding is fixed, this embedding can be treated as a layer of linear classifier, whose weights are obtained from the language embedding. This image-text contrastive objective can also be interpreted as using cross-entropy loss on a fixed readout layer that is initialized with the right language knowledge. This further validates the importance of language information for zero-shot adversarial robustness.

## A.6 ADAPTATION METHOD FORMULATION

**Token-Level Visual Prompts.** Token-level visual prompts adapt transformer-based models by appending tokens to the input token sequence. This is the most effective prompt method in our experiments, where we use this by default unless specified. Our visual prompts append additional tokens $P_k$ to the input sequence $\mathbf{x}$ of the vision transformer:

$$\mathbf{x} = [\mathbf{x}; P_0, P_1, ..., P_k] \tag{9}$$

The remaining transformer parameters and computations are kept the same as the original.

**Image-Level Visual Prompts.** Image-level visual prompts adapt transformer models by adding prompt to the input pixels. This is less effective way of adding prompt, as discussed in Figure 6a. Let the prompt token be $P$, and input image be $\mathbf{x}$ the visual prompt is added to the input image:

$$\mathbf{x} = \mathbf{x} + P \tag{10}$$

The remaining transformer parameters and computations are kept the same as the original.

**Finetuning.** This is the standard way of adapting a model, where all parameters of the model is updated with relatively small learning rate. In our experiments, we find learning rate of $1e-5$ achieves the best performance.

