# OpenReview forum: "Understanding Zero-shot Adversarial Robustness for Large-Scale Models"
_ICLR.cc/2023/Conference — ICLR 2023 poster_

### Official Review · Reviewer_8Zoa · 2022-10-25

**Confidence:** 4
**Correctness:** 4
**Technical Novelty And Significance:** 3
**Empirical Novelty And Significance:** 3
**Recommendation:** 8

**Clarity, Quality, Novelty And Reproducibility:**

# Clarity
-  For contributions: "our analysis provides useful lessons to understand the problem of zero-shot adversarial robustness."
	-  Their contributions would be easier to understand if the authors could list specific lessons instead (or highlight one or two most important lessons).
- I assume y_ij = -1 when i is not equal to j, but this is not explicitly mentioned.

# Novelty
- The problem setting (e.g. zero-shot adversarial robustness) and their approach is novel.

# Reproducibility
- They have enough details (e.g. Implementation Details section) to reproduce their results.

## Minor
- "finetuning has higher gains than VPT as more parameters are tuned." -> "Finetuning has..."
- Figure 7:  "By change(-ing) the interpolation ratio for the adapted CLIP and the vanilla CLIP,"

**Strength And Weaknesses:**

# Strength

- Zero-shot adversarial robustness is an important problem, given the fact that large-scale models are becoming some sort of infrastructure in practice.
- Their approach empirically performs well compared to other baselines.
- They even show that their approach doesn't require ground truth labels, and using psudo-labels (via CLIP image-to-text retrieval) is enough to attain similar performance.
- They have done comprehensive ablation study including:
	- The effect of text supervision in contrastive adversarial training.
	- The effect of visual prompt design (e.g. appending an additional token is better than adding a learnable noise to the raw input image.)
	- These ablation studies provide useful insights for future research in zero-shot adversarial robustness.

# Weaknesses

- No major weaknesses. See Clarify and Minor for small issues.

**Summary Of The Paper:**

CLIP has shown remarkable performance on zero-shot visual recognition. However, adversarial examples still greatly affect CLIP's performance. This work propose a text-guided contrastive adversarial training loss to adopt CLIP to attain adversarial robustness for the datasets that are not seen during adversarial training. They show that a naive adversarial training of CLIP on ImageNet achieves the best performance on ImageNet but fails on other classification datasets. Their approach, on the other hand, performs much better for zero-shot tasks, despite being slightly worse on ImageNet than the naive adversarial training. They evaluate their zero-shot performance on 15 image datasets, and perform comprehensive ablation studies of their method.

**Summary Of The Review:**

The problem of zero-shot adversarial robustness is important and this work illustrates how naive adversarial training cannot deal with zero-shot settings. Then they propose a new approach based on image-text embedding alignment to achieve superior zero-shot adversarial robustness. They also conduct a comprehensive set of ablation studies which yield useful insight for future research.

---

> ### Author Response · Authors · 2022-11-18
> **Thank you for your thoughtful review.**
>
> Thank you very much for your detailed reviews and suggestions for making the paper stronger.
>
> **Highlight Lessons in Contributions**
>
>  Following your suggestions, we rewrite the contribution paragraph in the introduction with highlighted lessons, which we marked as blue.
>
> **Missing Value for y_ij**
>
> Good catch. Sorry for missing this. We have updated the y_ij section. While your suggested one is also correct, our paper use y_ji=0 for i not equal to j, following the default contrastive learning setup.
>
> **Minor**
>
> Thank you for pointing this out. We have fixed it.

---

### Official Review · Reviewer_J4qy · 2022-10-25

**Confidence:** 4
**Correctness:** 3
**Technical Novelty And Significance:** 2
**Empirical Novelty And Significance:** 3
**Recommendation:** 3

**Clarity, Quality, Novelty And Reproducibility:**

The paper is readable, but some important details are missing. The novelty is limited as discussed in the Weaknesses. The reproducibility may be questionable due to some missing experimental details and that code is not provided.

**Strength And Weaknesses:**

[Strength]
1. The problem of zero-shot adversarial robustness is important in practice and has not been well explored.
2. Some of the results are interesting and may inspire future works on this problem, e.g., Figure 1 and Table 2.

[Weaknesses]
1. Technical novelty. The proposed TeCoA is nearly identical to: (1) first construct a zero-shot linear classifier head with the pre-defined text prompts for zero-shot classification, as done in (Wortsman et al., 2022), and then (2) perform vanilla adversarial training (using CE loss) with the classification head frozen. Hence the main technical contribution may be that the initialization (and/or freezing) of the classification head is important for adversarial fine-tuning.
2. Missing important experimental details. The missing implementation details include model architecture, training epochs and which visual prompt design is used. Besides, the generation of pseudo text labels for images (for results in Table 2) needs further explanation.
3. Robustness evaluation. PGD-100 may not be a reliable attack for robustness evaluation. It would be better to consider stronger attacks like AutoAttack (or at least the two APGD attacks used in AutoAttack) [1]. Besides, the main results in Table 1 are based on eps=1/255, which may be too small. Using larger eps like 4/255 as in [1] may be more convincing.

[1] Croce, Francesco, and Matthias Hein. "Reliable evaluation of adversarial robustness with an ensemble of diverse parameter-free attacks." International conference on machine learning. PMLR, 2020.

4. Explanation of abnormal results. As shown in Table 1, while FT (TeCoA) is the best on most datasets, LP (CE) and VPT (adv.) outperform other methods by a large margin on HateMemes and PACM. Is there any possible explanation?
5. Table 1 and Table 3 are inconsistent. Especially, there are two rows starting with "+VP (TeCoA)", which is confusing.

**Summary Of The Paper:**

This paper explores the problem of adapting large-scale pre-trained models for adversarially robust zero-shot classification. It is found that vanilla adversarial training on a single task may reduce the zero-shot capability of the pre-trained model. To improve the zero-shot adversarial robustness, a text-guided contrastive adversarial training (TeCoA) is proposed, which aligns the image embeddings of adversarial examples and the text embeddings of the standard prompts for zero-shot predictions. Experiments validate the effectiveness of TeCoA.

**Summary Of The Review:**

While this paper studies an interesting and not well-explored problem, and provides some experimental results that may promote the understanding of zero-shot adversarial robustness, there is a lack of insight and technical novelty. Besides, the missing details and weak robustness evaluation reduce the reliability of the results.

---

> ### Author Response · Authors · 2022-11-18
> **Thank you for your thoughtful review. We hope our responses address your questions.**
>
> Thank you very much for your helpful comments. We are glad that you find our problem important and novel, and our results are interesting. We are glad that you think our work may inspire future works on this important problem.
>
> **Technical novelty**
>
> Thank you for bringing up this point. The major novelty of our paper is the task: zero-shot adversarial robustness. As all the reviewers point out, given the fact that large-scale models are becoming some sort of infrastructure in practice, this is an important, yet unexplored problem to study. Our work is the first benchmark on this new task, showing promising results, which can inspire the vision field to continue to push this forward.
>
> Technically, our major finding is that using text knowledge in adversarial learning improves zero-shot transferability, while the standard no-text adversarial training does not. We thank the reviewer for interpreting this "adding text knowledge’’ way from a different angle: "adding text knowledge through a fixed classification that is initialized from the language model, and then using CE.’’ This interpretation further validates that the gain of zero-shot robustness is not due to the contrastive objective, but due to the language information. This conclusion is consistent with our empirical results by comparing TeCoA with baseline CoAdv and ImgCoAdv, which are ablation studies on the effect of language information. We are glad that the reviewer adds further intuition to this. We have added this perspective in the discussion of the multi-modal contrastive loss in our paper (Appendix A.5). We appreciate your suggestions to make our paper stronger.
>
> We would also mention that  (Wortsman et al., 2022) are different from ours because they add an extra linear head, while ours use the original CLIP and do not add any extra layer. We conduct contrastive learning directly on the representation layer of pretrained CLIP.
>
> **Experimental Details**
>
> Thank you for pointing this out! Following your suggestions, we add the model architecture (CLIP-B/32), training epochs (10 epochs), and prompt used (appended token-level prompt) in Section 4, Implementation details paragraph in blue.
>
> We add descriptions and algorithms for the generation of pseudo-text labels in Appendix A.3.5.
>
> **Evaluation on AutoAttack**
>
> Thank you for your suggestions to try AutoAttack. We cited it and applied our method to it and obtained an average of 36 points gain using fine-tuning, and 28 points gain using visual prompt tuning. Table 4 shows our Auto-Attack results and Appendix A.2 provides a discussion for the results. We find AutoAttack decreases the robustness of baseline CLIP by an average of 5.7 points, but slightly decreases ours by an average of 0.86, suggesting the robustness of our proposed method.
>
> **Evaluation on Larger Eps like 4/255**
>
> Thank you for your suggestions. We included the robust accuracy of eps=4/255 under PGD100 in Figure 5 in our initial submission. Following your suggestion, we add the robust accuracy of eps=4/255 under AutoAttack in Table 5. While zero-shot adversarial robustness is a  challenging task, our method improves robustness by an average of 9.1 points under this stronger attack.
>
> **Explanation of HateMemes and PCAM Results**
>
> Thank you for pointing this out. The HateMemes dataset (https://ai.facebook.com/blog/hateful-memes-challenge-and-data-set/) is a binary classification task for detecting hateful speech, which is a very different domain from the ImageNet classification. Since both LP(CE) and VPT (adv) only adapt a small number of parameters on the ImageNet set, the resulting model learns less and overfits less to the Image recognition task, and performs almost random guessing, resulting in an accuracy of 54%  on HateMemes, which is close to random guessing at 50%.
>
> PCAM is a binary classification for lymph nodes (medical image, https://github.com/basveeling/pcam), which is also a very different domain from ImageNet. Similar to HateMemes, adapting fewer parameters makes the model learn less and overfit less, resulting a 52.5% accuracy on PCAM which is close to random guessing at 50%.
>
> Thus, both datasets remain a big challenge for all existing zero-shot robust classification tasks. We believe the LP and VPT are not successfully attacked due to they might perform random guessing and not be sensitive to attack. We have added this discussion in Section A.4 in the appendix.
>
> **Name Inconsistency in Table 1 and Table 3.**
>
> Good catch. We have updated the name typo. Thanks for pointing out this.
>
> **Reproducibility**
>
> Thank you for your question. We upload our code in the supplementary for your and others' reference. We have included extra details in Section 4, and Appendix Section A.3. Our method is simple and effective, with few hyperparameters, which now should all be described in the paper. All of which should make our work reproducible by other researchers. We will release the models and cleaner code upon acceptance.

---

### Official Review · Reviewer_Uucq · 2022-10-25

**Confidence:** 3
**Correctness:** 4
**Technical Novelty And Significance:** 3
**Empirical Novelty And Significance:** 3
**Recommendation:** 8

**Clarity, Quality, Novelty And Reproducibility:**

The paper is very well written and should be reproducible from the description in the paper alone. I lack the background in zero-shot learning to judge the novelty of this work compared to prior work.

**Strength And Weaknesses:**

The paper is very well written and easy to follow: the motivation is clear, the objective and the training procedure are clearly defined. All experiments and ablations are clearly motivated, described, and reported as well. The results are promising. The provided intuition for why the proposed method helps while baselines fail makes intuitive sense.

The only part of the paper that I found to be not entirely clear and self-contained is the specifics of the image-to-image loss used in ImgCoAdv experiments.


**Summary Of The Paper:**

Authors are the first to show that zero-shot image recognition models built on top of CLIP are still susceptible to adversarial attacks, and that standard adversarial training objective, while effective at preventing adversarial attacks, destroys the rich image-language capability of CLIP, making the defended model useless for zero-shot recognition on different tasks/datasets. Authors propose TeCoA - a novel objective that takes the image-language nature of the model into account during adversarial training via visual prompt tuning (VPT) and contrastive learning. Authors show how the proposed objective interacts with various task adaptation techniques, including linear probing, full and partial finetuning, image and token-level prompting. More specifically, authors propose to perform an adversarial attack on the standard contrastive image-text alignment objective used in CLIP - an adversarial image perturbation aims to make the correct image-text pair to have lower cosine similarity then an incorrect one. Authors propose two ablations that verifies that observed gains indeed come from robustifying the vision-language model, and not from robustifying the vision backbone itself - contrastive alignment to one-hot vectors and an image-to-image loss. Authors show that when it comes to zero-shot adversarial robustness, the proposed approach beats the baseline adversarial training and both ablations, and that finetuning outperforms visual prompt tuning. Authors also investigate the effect of the dataset size, attack strength, prompt design, number of adapted parameters during fine-tuning, and the of pareto optimum the clear-vs-robust accuracy.

**Summary Of The Review:**


I think this is a great paper - it establishes an impactful problem, explores a simple solution in great detail with a lot of ablation experiments to verify their intuition, the paper is very well written.

---

> ### Author Response · Authors · 2022-11-18
> **Thank you for your thoughtful review.**
>
> Thank you for your detailed review and suggestions. We are glad that you find our paper clearly motivated, clearly written, and with promising results. We address your questions below:
>
> **Details for the image-to-image loss in ImgCoAdv experiments**
>
> Thank you for pointing this out. Following your suggestion, we add details for the ImgCoAdv algorithm in Section A.3.3.
>
> We also clarify here for your convenience. In contrast to our approach, ImgCoAdv does not use language information. On only image data, ImgCoAdv first creates an augmented image for each image as positive data, and then constructs a contrastive loss on the created data pairs. Then it optimizes the adversarial perturbation on the original image so that it maximizes the contrastive loss. Then ImgCoAdv adapts the model to minimize the contrastive loss on the attacked images and the augmented images, so the model performs robustly under attack.

---

### Official Review · Reviewer_eUU6 · 2022-10-25

**Confidence:** 2
**Correctness:** 3
**Technical Novelty And Significance:** 1
**Empirical Novelty And Significance:** 3
**Recommendation:** 6

**Clarity, Quality, Novelty And Reproducibility:**

Most explanations of the proposed method are clear and are presented with high quality. The technique is not novel and some implementation details are not given.

**Strength And Weaknesses:**

Strength:

The proposed problem is new and under-explored.

The paper conducts rich experiments to compare diverse adaptations methods and several possible training loss functions. Many datasets are employed to evaluate zero-shot generalization.

The proposed loss function is well motivated by the observation that text encoder is important to zero-shot transferability. Ablation study verifies that the proposed method using text embedding for contrastive learning is important.


Weakness:

Although the setting is new, the proposed loss function is a direct use of contrastive loss for two modalities. Moreover, the adaptation methods used are all existing techniques.

Implementation details are not sufficient. For example, it is unclear how exactly the adaptation methods are employed during training and what the objective functions of CoAdv or ImgCoAdv are. It would be better to give mathematical formulations of these methods in appendix.


**Summary Of The Paper:**

This paper studies the problem of zero-shot adversarial robustness: adapting pretrained large-scale vision-language models to unseen target tasks with high robust accuracies. With the conjecture that language encoder plays an important role in achieving good zero-shot generalization ability, a contrastive based adversarial training objective is proposed to contrast between image and text embeddings. Several adaptation methods are analyzed, and some interesting discoveries are made for the visual prompt tunning method.

**Summary Of The Review:**

This paper targets the problem of zero-shot adversarial robustness for CLIP, and it seems that it is a “first do A+B” type of work. Techniques used in the proposed methods are not new, while comprehensive experimental results show that the proposed method is effective through comparisons with reasonable baseline methods.

---

> ### Author Response · Authors · 2022-11-18
> **Thank you for your thoughtful review. We hope our responses address your concerns.**
>
> We appreciate your thoughtful review. We are glad that you find our proposed problem—zero-shot adversarial robustness—new, important, and should be explored. We are glad that you think our loss function is well motivated, and supported by rich experiments. We address the questions below:
>
> **The novelty of the proposed loss function**
>
> Thank you for your question. Our major contribution is that we study this important yet overlooked zero-shot robustness task. We agree with the reviewer that the image-text contrastive loss is standard. However, finding this loss— combined with adversarial training and the right adaptation method—can achieve high zero-shot robustness for the first time, is non-trivial and requires systematic analysis.  Our lesson on using language information in adversarial training for zero-shot robustness is significant and interesting. We believe this result should be published to the community.
>
> **Implementation details**
>
> Thank you for your suggestions. Following your suggestion, we add objective functions, descriptions, and algorithms for both CoAdv and ImgCoAdv in appendix Section A.3.2 and A.3.3. We also add a pointer to the appendix in the main paper. We add a mathematical formulation of the adaptation method including visual prompt in Appendix A.6.
>
> We hope we addressed your concerns. Please let us know if you’d like any further information.

---

### Public Comment · ~Muhammad_Maaz1 · 2023-03-12
**Release of Codebase**

Dear Authors,

Congratulations on the acceptance of your paper, which is a valuable contribution to understanding the zeroshot adversarial robustness of large-scale vision-language models.

I would like to inquire about the release date of a codebase that can reproduce the results of your paper. The GitHub link provided in the updated paper is not working, and the code attached to the supplemental is incomplete, lacking many dependencies and documentation. Thank you for your excellent work, and I look forward to hearing from you soon.

Best regards,
Maaz

---

### Decision · Program_Chairs · 2023-01-20

**Decision:**

Accept: poster

**Justification For Why Not Higher Score:**

Though the task is novel, the approach is clearly not as "professional" as those adv. defence works.

**Justification For Why Not Lower Score:**

A good starting point for the community to study the adv. attack of foundation models.

**Metareview: Summary, Strengths And Weaknesses:**

This paper presents a robust prompt tuning method TeCoA to counter the Adv attack in the context of using pre-trained vision-language models such as CLIP. Experimental results demonstrate good robustness.

Strengths:
A novel task in investigating the adv robustness in foundation models.

Weakness:
If we consider the paper as a fine-tuning paper, it is novel. However, if we consider it as an adv defence paper, it is over-simplified.

**Note From Pc:**

if the above contains the word "oral" or "spotlight" please see: "oral" presentation means -> notable-top-5% and "spotlight" means -> notable-top-25%. As stated in our emails, we are disassociating presentation type from AC recommendations

**Summary Of Ac-Reviewer Meeting:**

Reviewer J4qy was against this paper mainly due to the following three points, which were well-addressed in rebuttal: 1) non-novel technical contribution (which was however not the claim of the paper). 2) missing implementation details (which were provided in the revision), and 3) some confusing experimental results (which turned out to be some typos). AC read the paper, reviews, and rebuttal, and found that the paper's main weakness is indeed the technical contribution; however, considering that the technical contribution is not the main claim of the paper, AC agreed with the other three positive reviewers and urged Reviewer J4qy to respond the authors. Unfortunately, the reviewer failed to do so and AC eventually recommended acceptance.